# Circ_0011446 Regulates Intramuscular Adipocyte Differentiation in Goats via the miR-27a-5p/FAM49B Axis

**DOI:** 10.3390/ijms26052294

**Published:** 2025-03-05

**Authors:** Jian-Mei Wang, Jin-Shi Lv, Ke-Han Liu, Yan-Yan Li, Jiang-Jiang Zhu, Yan Xiong, Yong Wang, Ya-Qiu Lin

**Affiliations:** 1Key Laboratory of Qinghai-Tibetan Plateau Animal Genetic Resource Reservation and Utilization, Ministry of Education, Southwest Minzu University, Chengdu 610041, China; wanganmei23@163.com (J.-M.W.); 18877548381@163.com (J.-S.L.); LKH2485387606@126.com (K.-H.L.); liyanyan@swun.edu.cn (Y.-Y.L.); zhujiang4656@hotmail.com (J.-J.Z.); xiongyan0910@126.com (Y.X.); 2Key Laboratory of Qinghai-Tibetan Plateau Animal Genetic Resource Reservation and Exploitation of Sichuan Province, Southwest Minzu University, Chengdu 610041, China; 3College of Animal & Veterinary Science, Southwest Minzu University, Chengdu 610041, China

**Keywords:** goat, intramuscular adipocyte differentiation, circ_0011446, miR-27a-5p, FAM49B

## Abstract

Intramuscular fat (IMF), or marbling, is a critical indicator of goat meat quality. Non-coding RNAs play a key role in the formation and deposition of IMF in vertebrates by regulating genes involved in its synthesis, degradation, and transport. The competing endogenous RNA (ceRNA) hypothesis identifies circular RNAs (circRNAs) as natural “sponges” for microRNAs (miRNAs). However, the precise mechanisms of circRNAs in goat IMF remain poorly understood. In the current study, we utilized existing sequencing data to construct a ceRNA regulatory network associated with intramuscular adipogenesis and fat deposition in goats. Our goal was to elucidate the post-transcriptional regulatory mechanism of family with sequence similarity 49 member B (FAM49B). Functionally, FAM49B was found to inhibit the differentiation of intramuscular preadipocytes and to directly interact with miR-27a-5p. Mechanistically, dual-luciferase reporter assays and quantitative real-time PCR (qRT-PCR) confirmed the interaction between circ0011446 and miR-27a-5p. Circ0011446 enhanced the expression of FAM49B mRNA and protein through post-transcriptional regulation. As a ceRNA, circ0011446 competitively binds miR-27a-5p, preventing miR-27a-5p from degrading FAM49B. In conclusion, our findings demonstrate that circ0011446 suppresses goat adipogenic differentiation of intramuscular preadipocytes by regulating the expression of the downstream target gene FAM49B through miR-27a-5p sequestration. This study provides a reference for goat meat quality or livestock breeding.

## 1. Introduction

Goat meat is one of the most widely consumed meats globally. In contrast to beef and pork, it serves as a significant source of high-quality protein, beneficial fats, and low-calorie intramuscular fat, while also containing lower levels of saturated fats [1]. As such, goat meat plays a vital role in human nutrition. Intramuscular fat (IMF) plays a crucial role in the flavor, tenderness, and juiciness of goat meat; an optimal IMF content helps improve meat quality and flavor [2,3]. IMF refers to fat stored within the muscle, mainly in the epimysium, perimysium, and endomysium, where it disrupts intramuscular connective tissue, enhancing meat tenderness [4]. Studies indicated that fat deposition was mainly influenced by the proliferation, differentiation, and maturation of preadipocytes [5,6]. The number and aggregation capacity of preadipocytes were key factors determining the IMF content [7], with adipocyte differentiation playing a decisive role in this process. The formation of IMF is a complex biological process shaped by genetic, environmental, nutritional, and disease factors, and is regulated by genes, transcription factors, non-coding RNAs, and signaling pathways [8,9]. Understanding these complex pathways is crucial for manipulating the IMF content in meat products to meet consumer preferences and market demands.

In recent years, whole transcriptome technology has been widely applied in the livestock industry, leading to the identification of numerous non-coding RNAs (ncRNAs) as key regulators of IMF deposition. Circular RNAs (circRNAs), a novel class of endogenous ncRNAs with a covalently closed loop structure, are found in the transcriptomes of various species and tissues [10]. Unlike linear RNAs, circRNAs exhibit higher structural stability and conservation. While most circRNAs are located in the cytoplasm, a small subset, derived from introns, resides in the nucleus [11]. CircRNAs are classified into three types based on their genomic origin and biogenesis: intronic, exonic, and exon–intron circRNAs [12,13,14]. Studies have shown that circRNAs function as molecular sponges for microRNAs (miRNAs), regulating miRNA expression and their target genes, thus influencing various biological processes [15]. For instance, circFut10 inhibits adipocyte differentiation by sponging miR-7 in cattle, while circPPARγ promotes adipocyte differentiation through binding miR-92a-3p and YinYang 1 (YY1) [16]. Additionally, circSAMD4A regulates adipogenesis in porcine preadipocytes via the miR-138-5p/EZH2 pathway [17]. These findings highlight the crucial role of circRNAs in adipogenesis. Despite the fact that numerous circRNAs related to adipogenesis have been identified, their specific roles and mechanisms in goat adipogenesis remain to be further explored. 

In our previous study, we systematically identified circular RNAs (circRNAs) associated with IMF differentiation and performed circRNA sequencing (circRNA-seq) on goat intramuscular adipocytes before and after induced differentiation [18]. Based on these results, this study continues to use Jianzhou Da’er goats as a model, focusing on circ_0011446, which shows significant expression differences before and after the differentiation of goat intramuscular adipocytes. The findings further validated the role of circ_0011446 in goat adipogenesis, providing new insights into the biological functions and regulatory mechanisms of IMF deposition in goats. This has important implications for enhancing the IMF content in goat muscle.

## 2. Results

### 2.1. Circ_0011446 Identification

Circ_0011446 was derived from exons 4 to 7 of the MYPN gene located on chromosome 28 of goats (Figure 1A). Sanger sequencing confirmed the back-splice junction sequences and cyclization site of circ_0011446 (Figure 1A). Amplification using both divergent and convergent primers in RNase R (+) and RNase R (−) groups confirmed that circ_0011446 possesses a circular structure (Figure 1B). qRT-PCR analysis was performed to detect the expression levels of circ_0011446 at different time points during intramuscular adipocyte differentiation in goats (Figure 1C). The results shows that the expression of circ_0011446 decreased in the early stage to 24 h and stabilized from 24 h to 72 h, then increased to 96 h, and decreased again until the end (120 h).

### 2.2. Circ_0011446 Negative Regulator Adipogenic Differentiation of Goat Preadipocytes

We observed that the expression of circ_0011446 was significantly lower during the 0–72 h period of goat intramuscular adipocyte differentiation compared to the baseline (0 h), suggesting its potential role in inhibiting the early differentiation of intramuscular preadipocytes (IMPAs). Thus, we overexpressed (OE) circ_0011446 in goat intramuscular adipocytes using adenovirus. After 72 h, qRT-PCR analysis showed a significant upregulation of circ_0011446 expression (*p* < 0.01, Figure 2A). Bodipy and Oil Red O staining revealed that the overexpression of circ_0011446 inhibited lipid droplet accumulation in goat intramuscular adipocytes (Figure 2B,C). The expression of adipogenic marker genes C/EBPα, C/EBPβ, and AP2 was significantly downregulated (*p* < 0.05, Figure 2D), while PPARγ and SREBP1 expression showed no significant differences (*p* > 0.05, Figure 2D). Meanwhile, we performed a circ_0011446 knockdown, which resulted in a 57% reduction in circ_0011446 expression after 72 h (*p* < 0.01, Figure 2E). Bodipy and Oil Red O staining indicated that the knockdown of circ_0011446 promoted lipid droplet accumulation in adipocytes (Figure 2F,G). The expression of adipogenic marker genes PPARγ, C/EBPα, and SREBP1 was significantly upregulated (*p* < 0.01, Figure 2H), and C/EBPβ expression was significantly downregulated (*p* < 0.05, Figure 2H). In conclusion, these results demonstrated the that overexpression and knockdown of circ_0011446 inhibited and promoted goat intramuscular adipocyte differentiation, respectively.

### 2.3. Circ_0011446 Serves as a miR-27a-5p Sponge

Nuclear–cytoplasmic separation and FISH analyses revealed that circ_0011446 was predominantly localized in the cytoplasm, consistent with the typical localization of eCircRNAs, suggesting its potential role as a competitive endogenous RNA (ceRNA) (Figure 3A,B). Next, we used RNAHybrid to predict the potential miRNAs of circ_0011446 and intersected them with miRNAs previously reported to be differentially expressed during goat intramuscular fat (IMF) differentiation [19]. We selected miR-27a-5p, which had the lowest free energy, for further study (Figure 3C). A lower minimum free energy (mfe) indicates a stronger binding between circRNAs and miRNAs [19]. Our results demonstrated that circ_0011446 negatively regulated miR-27a-5p, as the overexpression of circ_0011446 significantly inhibited miR-27a-5p expression (*p* < 0.01, Figure 3D). Furthermore, the knockdown of circ_0011446 significantly increased its expression (*p* < 0.05, Figure 3E). Finally, to validate the targeting interaction between circ_0011446 and miR-27a-5p, a wild vector containing binding sites and a mutant vector without binding sites were constructed (Figure 3F). Intramuscular adipocyte cells were co-transfected with dual luciferase reporter vectors along with miR-27a-5p mimics or NC mimics. As anticipated, the dual luciferase reporter gene assay demonstrated that miR-27a-5p mimics markedly inhibited the luciferase activity of the wild-type (WT) vector, while co-transfection with the mutant vector showed no significant inhibitory effect (*p* < 0.01, Figure 3F). These findings strongly support the direct targeting of the circ_0011446 gene by miR-27a-5p.

### 2.4. miR-27a-5p Positively Modulates Adipogenic Differentiation in Goat Preadipocytes

Furthermore, we overexpressed miR-27a-5p in goat intramuscular adipocytes using mimics. qRT-PCR analysis showed a significant upregulation of miR-27a-5p expression after 72 h (*p* < 0.01, Figure 4A). Morphological observations indicated that the mimics promoted lipid droplet accumulation in IMPA (Figure 4B,C) and upregulated the expression of PPARγ, C/EBPα, C/EBPβ, AP2, and SREBP1 (*p* < 0.05, Figure 4D). Concurrently, we performed a miR-27a-5p knockdown, which resulted in a significant downregulation of miR-27a-5p expression (*p* < 0.01, Figure 4E). Morphological observations showed that the knockdown inhibited lipid droplet accumulation in IMPA (Figure 4F,G), and downregulated the expression of PPARγ, C/EBPα, C/EBPβ, and SREBP1 (*p* < 0.05, Figure 4H). Together, these results suggested that the overexpression and knockdown of miR-27a-5p promoted and inhibited goat intramuscular lipid accumulation, respectively.

### 2.5. FAM49B Was the Direct Target of miR-27a-5p

The potential interaction between miR-27a-5p and FAM49B was predicted using online tools such as TargetScan, MIRDB, and Starbase. These results were then compared with genes previously reported to be differentially expressed during goat intramuscular fat (IMF) differentiation [19] (Figure 5A). We identified 10 potential target genes of miR-27a-5p, with FAM49B showing the strongest binding affinity and was associated with lipid deposition. We speculated that miR-27a-5p may bind to the 3′ UTR of FAM49B. To confirm this interaction, we found that overexpression of miR-27a-5p in intramuscular preadipocytes significantly reduced FAM49B mRNA expression. In contrast, knockdown of miR-27a-5p led to a significant increase in FAM49B mRNA expression (Figure 5B,C). Dual-luciferase reporter assays also showed that miR-27a-5p significantly reduced the luciferase activity of the FAM49B 3′ UTR WT (*p* < 0.01, Figure 5D). These results strongly support the direct targeting of the FAM49B gene by miR-27a-5p.

### 2.6. FAM49B Negative Regulator Adipogenic Differentiation of Goat Preadipocytes

To investigate the impact of family with sequence similarity 49 member B (FAM49B) on the differentiation of intramuscular preadipocytes, we successfully overexpressed FAM49B in the cells (Figure 6A). Subsequently, we observed that FAM49B overexpression reduced goat intramuscular lipid droplet accumulation (Figure 6B,C). qRT-PCR analysis was performed to assess the effect of FAM49B overexpression on intramuscular adipogenesis at the molecular level goat intramuscular lipid droplet accumulation. The data showed that overexpression significantly suppressed the mRNA levels of PPARγ, C/EBPα, C/EBPβ, and SREBP1 (Figure 6D). Conversely, knockdown of FAM49B promoted goat intramuscular adipocyte lipid droplets accumulation, with an upregulation of the mRNA levels of adipogenesis marker genes C/EBPα, C/EBPβ, SREBP1, and AP2 in siRNA1-treated cells. Thus, these data indicated that FAM49B loss of function promotes goat intramuscular adipogenesis.

## 3. Discussion

Jianzhou Da’er goats, China’s second-largest meat goat breed, are renowned for their tender meat, excellent flavor, unique taste, and high nutritional value, enjoying widespread popularity [20]. Thus, understanding their muscle lipid traits aids in developing and utilizing meat goat breeds, and protecting resources. circRNAs, once dismissed as abnormal RNA splicing products or pathogen structures due to their low abundance and expressive richness [21,22], have recently garnered attention. Studies reveal their ubiquitous presence in archaea, hinting at significant biological functions [23]. Advances in genome research have uncovered circRNAs in diverse eukaryotic cells and tissues [24,25]. We previously reported the differential expression of circRNAs in goat intramuscular fat (IMF) before and after differentiation [18]. A novel candidate circRNA, circ_0011446, exhibited differential expression and was identified as of interest. This circRNA is prominently expressed in intramuscular pre-adipocytes and undergoes downregulation as adipocytes differentiate, indicating a potential pivotal role in animal fat deposition.

Adipogenesis constitutes a process involving adipocyte proliferation, differentiation, and ultimate formation, all meticulously regulated by a suite of adipogenic factors. These regulatory elements orchestrate adipocyte exit from the cell cycle, induce the expression of adipocyte-specific genes, and suppress the transcription of other cell- or tissue-specific genes. Recently, circular RNAs (circRNAs) have been implicated in adipogenesis, exhibiting functional roles [26,27,28]. In our study, we discovered that circ_0011446, derived from the MYPN gene, inhibits differentiation in goat preadipocytes, potentially influencing the variability in intramuscular adipocyte deposition rates in goats. To investigate this, we suppressed the expression of circ_0011446 and noted an elevation in lipid droplet accumulation along with an upregulation of genes marking adipogenic differentiation. This indicated that the silencing of circ_0011446 facilitated adipocyte differentiation. In contrast, when circ_0011446 was overexpressed, adipogenesis was inhibited, as demonstrated by the marked downregulation of crucial adipocyte differentiation markers such as C/EBPα, C/EBPβ, and AP2. Based on these findings, we hypothesize that circ_0011446 functions as a negative regulator of fat deposition in goats. PPARγ, a pivotal regulator predominantly expressed in adipose tissue, is renowned for its role in facilitating fatty acid transport and lipid synthesis. It serves as a crucial inducer of adipocyte differentiation and fat metabolism [29]. Typically, PPARγ enhances intramuscular fat (IMF) deposition by upregulating genes associated with fat synthesis and downregulating those that inhibit fat production. Additionally, adipogenic differentiation marker genes, including C/EBPα, C/EBPβ, and AP2, play essential roles in the adipogenesis process. Zuo et al. [30] established that C/EBPβ initiates adipocyte differentiation by activating PPARγ, serving as a crucial pathway. Its absence impedes differentiation. However, our study results indicate that interference with circ_0011446 significantly downregulates C/EBPβ, a fat formation marker. This may occur during the transition from precursor to mature adipocytes, where high C/EBPβ levels are needed to activate PPARγ and initiate differentiation. Circ_0011446 interference disrupts this mechanism, reducing C/EBPβ expression.

There is a great deal of research demonstrating that circRNAs can serve as ceRNAs, protecting mRNAs by functioning as molecular sponges for miRNAs. This mechanism modulates the derepression of miRNA targets, thereby introducing an additional layer of post-transcriptional regulation. Given that circ_0011446 is an ecircRNA, elucidating its cellular localization is pivotal for deciphering its underlying mechanism. Our study initially validated the primary localization of circ_0011446 in the cytoplasm, aligning with the usual distribution pattern of ecircRNAs, which predominantly serve as competing endogenous RNAs (ceRNAs) within the cytoplasm [31]. A great deal of research has demonstrated circRNAs’ function as miRNA sponges, regulating fat deposition in animals [32]. MicroRNAs (miRNAs), short non-coding RNAs, regulate gene expression across diverse biological processes [33,34]. Notably, miRNAs have been implicated in adipogenesis and metabolic regulation [35,36]. For example, miR-669a-5p promotes adipocyte differentiation and induces the browning of preadipocytes [37], while the loss of miR-22 limits white adipose tissue expansion and activates brown adipose tissue, reducing fat accumulation caused by a high-fat diet [38]. However, research on miR-27a-5p has primarily focused on cancer, apoptosis, and inflammatory damage [39,40,41,42], with a limited exploration of its role in lipid metabolism. In this paper, the interaction of miR-27a-5p with circ_0011446 was further validated by RNAhybrid and the dual-luciferase reporter assay. Functionally, we observed an increase in miR-27a-5p expression upon the overexpression of circ_0011446 and a decrease upon its silencing, suggesting that circ_0011446 negatively regulates miR-27a-5p. To investigate the mechanism by which circ_0011446 regulates intramuscular preadipocyte (IMPA) differentiation, we manipulated miR-27a-5p levels in goat intramuscular preadipocytes by overexpressing and inhibiting it. Our results indicated that miR-27a-5p mimics facilitated adipocyte differentiation, whereas its inhibition hindered differentiation, implying that miR-27a-5p functions as a positive regulator of IMPA differentiation. Furthermore, a study conducted by Yang et al. [43] revealed that the calcium-sensing receptor (CASR) is a direct target of miR-27a-5p, and that miR-27a-5p may promote fat deposition in castrated bulls by targeting CASR.

Previous studies have consistently shown that miRNAs regulate gene expression post-transcriptionally, primarily through translation inhibition, mRNA degradation, or silencing of target transcripts [44,45,46]. In our previous RNA-seq analysis of goat intramuscular adipocytes, we identified miR-27a-5p as differentially expressed between goat intramuscular preadipocytes and adipocytes. Furthermore, we pinpointed FAM49B as a potential target gene of miR-27a-5p using TargetScan, MIRDB, and Starbase databases, and validated their interaction through dual-luciferase reporter assays. These results suggest that miR-27a-5p may directly impact the differentiation of goat intramuscular preadipocytes by targeting the 3′ UTR of FAM49B. FAM49B (family with sequence similarity 49 member B) is a gene highly conserved across mammalian species [47]. Intriguingly, this function aligns with the role of circ_0011446 but contrasts with the effect of miR-27a-5p, positioning FAM49B as a downstream effector in the circ_0011446/miR-27a-5p regulatory axis.

In conclusion, our in vitro results demonstrated that circ_0011446 regulates the downstream target gene FAM49B by sponging miR-27a-5p, thereby suppressing intramuscular fat breakdown and reducing IMF deposition. However, we recognize that the lack of in vivo experiments is a limitation of this study. The absence of validation in a living model restricts our ability to fully understand how circ_0011446 modulates the proliferation and differentiation of goat intramuscular preadipocytes via the miR-27a-5p/FAM49B pathway. Despite this limitation, our findings offer novel insights into the mechanisms controlling IMF deposition in meat goat breeds and provide a foundation for future investigations. We aim to build on these results by conducting in vivo studies to further validate and expand our understanding of this regulatory axis.

## 4. Materials and Methods

### 4.1. Cell Isolation and Culture of Primary Goat Intramuscular Preadipocytes

The animal experiments were approved by the Animal Experimental Ethics Committee of Southwest Minzu University (No.2020086, 2020). For this study, seven-day-old Jianzhou Daer goats (*n* = 3) were obtained from Sichuan Tiandi Goat Bioengineering Co., Ltd., located in Chengdu, China. At 7 days old, intramuscular preadipocytes are in an early developmental stage, suitable for studying adipogenesis and cell differentiation due to their high proliferation and differentiation capacity, and the pre-weaning stage of animals ensures a standardized starting material less influenced by external factors. All goats were fed and managed under standard environmental and nutritional conditions. Preadipocytes were isolated from three 7-day-old goats adipose tissues and combined to minimize individual variation, as described in a previous study [48]. Briefly, the longissimus dorsi muscle adipose tissues were collected under aseptic conditions, and then digested with collagenase type II (Gibco, USA) for 90 min at 37°C. Subsequently, the samples were centrifuged and the supernatant containing the goat preadipocytes was collected. Goat preadipocytes were cultured in complete medium (DMEM containing 10% fetal bovine serum) (Gibco, Grand Island, NY, USA). When the cell confluence reached more than 90%, cell differentiation was induced with the differentiation medium (50 µmol·L^−1^ oleic acid).

### 4.2. Adenovirus Generation and Cell Infection

The adenoviruses overexpressing circ_0011446 or FAM49B were constructed and packaged by Shanghai Hanbio (China), with a final titer of 3.16 × 10^10^ PFU/mL. To overexpress circ_0011446 (OE-circ_0011446) or FAM49B (OE-FAM49B), cells were transfected with pEGFP-N1-circ_0011446 or pEGFP-N1-FAM49B plasmids using Lipofectamine 3000 (Thermo Fisher Scientific, Waltham, MA, USA) when the cells reached approximately 70% confluence in complete medium. A negative control (NC) for overexpression was included. Six hours after transfection, when the cell confluence reached over 90%, the cells were switched to a differentiation medium for further culture. After 72 h, GFP fluorescence was observed, and the cells were collected for subsequent analysis.

### 4.3. Chemical Synthesis of siRNA and Cell Transfection

The knockdown circ_0011446 (pGPU6-circ_0011446) was designed and synthesized by RiboBio (Guangzhou, China). miR-27a-5p mimics, negative control (NC) mimics, an miR-27a-5p inhibitor, and NC inhibitor were synthesized by Jima Pharmaceutical Technology (Shanghai, China) (Table 1). The above RNAs, FAM49B-wild (ligated the 3′-UTR of FAM49B) and pEGFP-N1 plasmids were transfected by Lipofectamine™ 3000 Reagent (Invitrogen, L3000015, USA), according to the instructions of the manufacturer. Then, adipogenic differentiation was performed on the preadipocyte cells, and the cells were analyzed adipogenic marker expression using qRT-PCR along with lipid accumulation assessed by Bodipy and Oil Red O staining at day 3 after induction.

### 4.4. Validation of circRNA

The validation of circRNAs was carried out through PCR using both divergent and convergent primers. To verify the junction sequences of circRNAs, PCR products obtained with divergent primers were gel-purified and sent for Sanger sequencing at Sangon Biotech (Shanghai, China) Co. To assess the sensitivity of circRNAs to RNase R, PCR was performed on RNA samples with and without RNase R treatment. The primers used for circRNA validation are listed in Table 2.

### 4.5. RNA Extraction and Quantitative Real-Time PCR (qRT-PCR) 

Total RNA was extracted from various tissues using TRIzol (TaKaRa, Tokyo, Japan). For qRT-PCR analysis, the cDNAs of mRNA, circRNA, and miRNA were synthesized using the PrimeScript RT Reagent Kit (Takara, Shiga Prefecture, Japan) for mRNA and circRNA, and the miRNA 1st Strand cDNA Synthesis Kit (by stem-loop) Vazyme, Nanjing, China) for miRNA, respectively. qRT-PCR was performed using SYBR Green Master Mix (TaKaRa, Tokyo, Japan) for mRNA and circRNA, and the miRNA Universal SYBR qPCR Master Mix Kit (Vazyme, Nanjing, China) for miRNA. *UXT*, *U6*, and *GAPDH* served as internal reference genes for normalization [49]. Widely used in adipocyte differentiation studies, *UXT* demonstrates consistent expression levels. Commonly employed for normalizing small non-coding RNAs, such as microRNAs, *U6* exhibits stable expression in cellular contexts. Extensively validated in similar studies, *GAPDH* is a well-established housekeeping gene with stable expression in multiple cell types, including preadipocytes. Ensuring accurate normalization and the reliable quantification of gene expression, these reference genes were utilized in our experiments. The quantification of gene expression was carried out using the comparative threshold cycle (2^−ΔΔCT^) method [50]. The experiment was repeated three times. The primers used for qRT-PCR are listed in Table 3.

### 4.6. Nuclear–Cytoplasmic Isolation and Fluorescence In Situ Hybridization (FISH)

RNA from the nucleus and cytoplasm of IMPAs was extracted using the PARI^TM^ Kit (Invitrogen, Carlsbad, CA, USA). In brief, 1 × 10^7^ cells were centrifuged to collect a cell pellet. The cytoplasmic fraction was isolated from the supernatant after adding Cell Fraction Buffer to the pellet. The nucleus fraction was obtained by adding Cell Disruption Buffer to the remaining pellet.

The Cy3-labeled probe specific to circ_0011446 was designed and synthesized by RiboBio Co., Ltd. (Guangzhou, China). The cellular localization of circ_0011446 was determined using a FISH kit (RiboBio, Guangzhou, China).

### 4.7. Bodipy and Oil Red O Staining

Goat intramuscular adipocytes were washed twice with PBS and fixed in 4% formaldehyde for 15 min at room temperature. The cells were then incubated with the Oil Red O working solution, which contained 6 mL of Oil Red O stock solution (5 g/L in isopropanol) and 4 mL of ddH2O, for 30 min. After staining, the cells were washed with 60% isopropanol in PBS and imaged. The Oil Red O dye was extracted with 100% isopropanol, and the signal was quantified by measuring the optical density at 490 nm (OD 490).

Goat intramuscular adipocytes were isolated from the induction medium, incubated with 2 µM BODIPY^TM^ 493/503 (Thermo Fisher Scientific, Waltham, MA, USA, D3922), and diluted in IM for 1 h. The cells were then washed three times with PBS, refreshed with induction medium, and imaged. Fluorescent images were captured using an Olympus TH4–200 microscope (Olympus, Tokyo, Japan) and analyzed by the ImageJ 1.54k tool (NIH, Bethesda, MD, USA).

### 4.8. Dual-Luciferase Reporter Assay

The dual-luciferase reporter assay was conducted as previously described [51]. Wild-type and mutant plasmids of circ_0011446 and FAM49B were constructed using the pmirGLO vector (GenePharma, Shanghai, China). The plasmids were designated pmirGLO-circ_0011446 WT, pmirGLO-circ_0011446 MT, pmirGLO-FAM49B 3′ WT, and pmirGLO-FAM49B 3′ MT. These plasmids were transfected into intramuscular adipocytes at approximately 70% confluence. After 48 h, luciferase activity was measured using the Dual Luciferase Reporter Assay Kit (Vazyme, Nanjing, China), and the firefly-to-Renilla luciferase ratio was calculated.

### 4.9. Statistical Analysis

Data are presented as means ± SEM. Statistical comparisons were performed using the Student’s *t*-test or one-way ANOVA with SPSS 20.0 software (SPSS Science, Chicago, IL, USA). *p* values < 0.05 were considered statistically significant. Results are shown as the mean ± SEM and the data are representative of three biological and two technical replicates.

## 5. Conclusions

In conclusion, our findings demonstrate that circ_0011446 regulates FAM49B expression by acting as a sponge for miR-27a-5p, thereby influencing the differentiation of intramuscular preadipocytes and modulating intramuscular fat formation and deposition in goats (Figure 7). These results provide a strong foundation for understanding the mechanisms and regulatory networks involved in lipogenesis and will serve as a valuable resource for future research in this field.

## Figures and Tables

**Figure 1 ijms-26-02294-f001:**
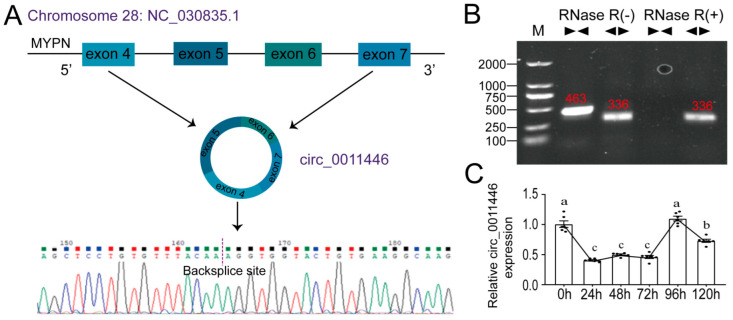
Identification of circ_0011446. (**A**). Cyclization diagram of circ_0011446 and sequence diagram of the back-splicing site analysis. (**B**). circ_0011446 ring-forming identification. M is the marker; RNase R (+) and RNase R (−) represent whether there is RNase R digestion; 
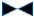
 and 
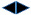
 represent the convergent prime and divergent prime, respectively. (**C**). Relative circ_0011446 expression during goat intramuscular adipocyte differentiation. Different lowercase letters indicate significant differences between groups according to one-way ANOVA (*p* < 0.01).

**Figure 2 ijms-26-02294-f002:**
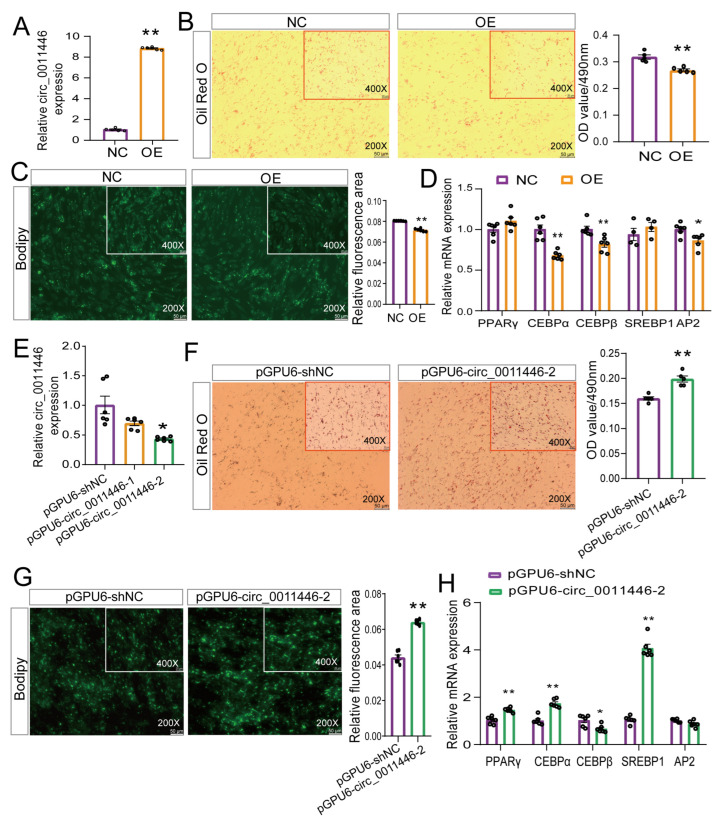
circ_0011446 inhibits adipogenic differentiation in goat. (**A**). Overexpression efficiency of circ_0011446. (**B**,**F**). Images of mature intramuscular adipocytes stained with Oil Red O (**left panel**) and OD value determination (**right panel**) following circ_0011446 overexpression (**B**) and knockdown (**F**), respectively. Oil Red O staining signal was quantified by absorbance at 490 nm. (**C**,**G**). Staining of mature intramuscular adipocytes with Bodipy (**left panel**) and fluorescence area quantification (**right panel**) after overexpression (**C**) and knockdown (**G**) of circ_0011446, respectively. (**D**,**H**). mRNA expression levels of adipogenic marker genes following circ_0011446 overexpression (**D**) and knockdown (**H**), respectively. (**E**). Interference efficiency of circ_0011446. All data are presented as mean ± SEM, * *p* < 0.05, ** *p* < 0.01.

**Figure 3 ijms-26-02294-f003:**
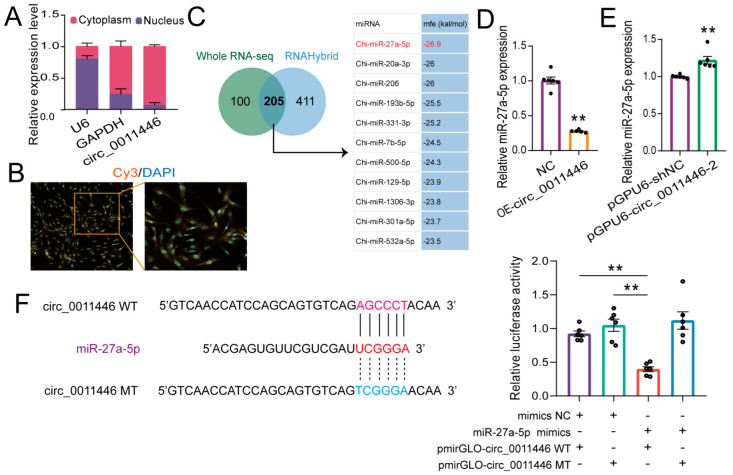
circ_0011446 acts as a miR-27a-5p sponge. (**A**). Nucleocytoplasmic separation demonstrated the predominant cytoplasmic localization of circ_0011446. (**B**). Subcellular localization of circ_0011446 shown by FISH. (**C**). Potential miRNA binding sites of circ_0009659 predicted by RNAHybrid. (**D**,**E**). Expression changes of miR-27a-5p after overexpression (**D**) or knockdown (**E**) of circ_0011446. (**F**). Dual-luciferase reporter assay results showed that miR-27a-5p targets circ_0011446. All data are presented as mean ± SEM, ** *p* < 0.01.

**Figure 4 ijms-26-02294-f004:**
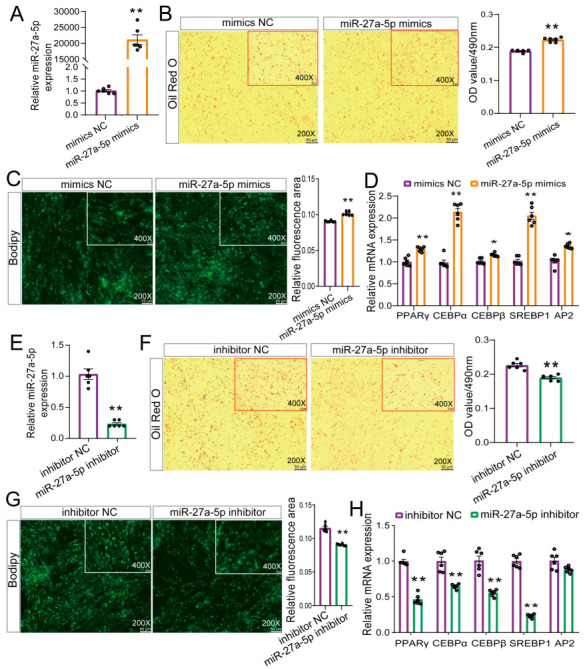
Effect of miR-27a-5p on differentiation of goat intramuscular preadipocytes. (**A**,**E**). Overexpression (**A**) or knockdown (**E**) efficiency of miR-27a-5p detected by qRT-PCR. (**B**,**F**). The effect of miR-27a-5p overexpression (**B**) or knockdown (**F**) on adipogenic differentiation was quantitatively assessed by Oil Red O staining, with the OD value of Oil Red O dye at 490 nm measured. (**C**,**G**). Bodipy staining analysis of miR-27a-5p overexpression (**C**) or knockdown (**G**) in intramuscular preadipocytes. (**D**,**H**). The effect of miR-27a-5p overexpression (**D**) or knockdown (**H**) on adipogenic marker gene expression was detected by qRT-PCR. * *p* < 0.05, ** *p* < 0.01, compared to that of NC.

**Figure 5 ijms-26-02294-f005:**
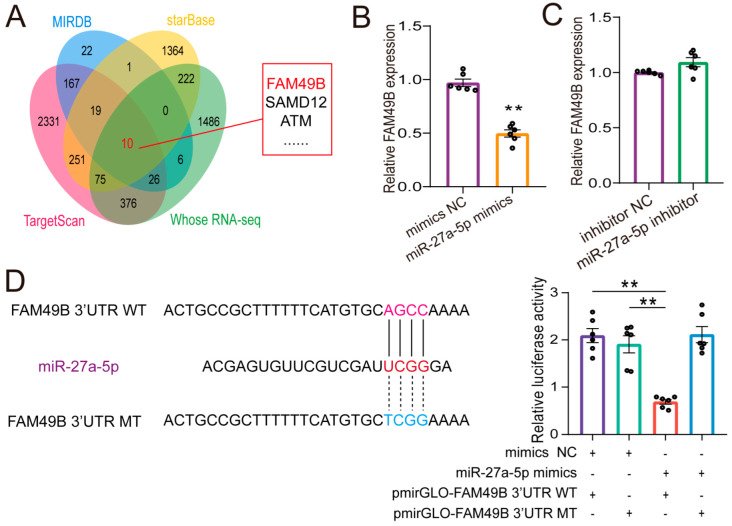
Prediction and validation of the targeting relationship between miR-27a-5p and FAM49B. (**A**). TargetScan, MIRDB, and Starbase predicted miR-27a-5p targeting FAM49B. (**B**,**C**). The effect of miR-27a-5p overexpression (**B**) and knockdown (**C**) on FAM49B expression was detected by qRT-PCR. (**D**). A dual-luciferase reporter assay verified the targeting relationship between miR-27a-5p and FAM49B (**Right panel**), with construction of wild-type and mutant vectors (**Left panel**). The data were shown as the mean ± SEM (*n* = 3) (** *p* < 0.01).

**Figure 6 ijms-26-02294-f006:**
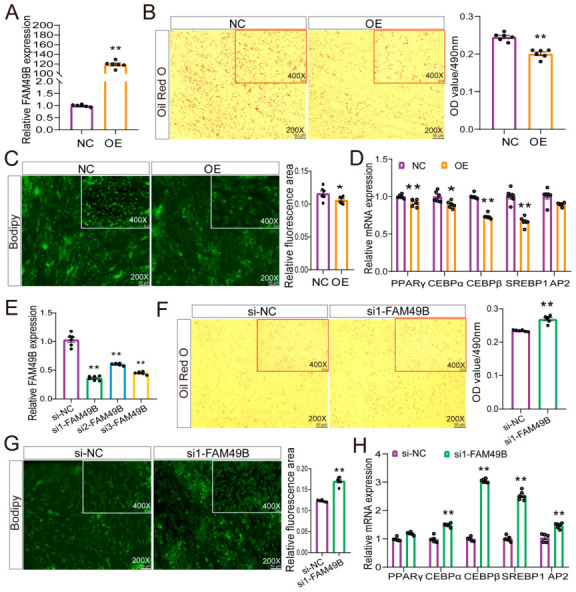
Role of FAM49B in goat intramuscular preadipocyte differentiation. (**A**,**E**). FAM49B overexpression (**A**) or knockdown (**E**) efficiency. (**B**,**F**). Oil Red O staining of mature adipocytes (**left**) and OD value at 490 nm (**right**) after FAM49B overexpression (**B**) or knockdown (**F**). (**C**,**G**). Bodipy staining of mature adipocytes (left) and fluorescence area quantification (right) after FAM49B overexpression (**C**) or knockdown (**G**). (**D**,**H**). mRNA levels of adipogenic markers after FAM49B overexpression (**D**) or knockdown (**H**). Data are presented as mean ± SEM, * *p* < 0.05, ** *p* < 0.01.

**Figure 7 ijms-26-02294-f007:**
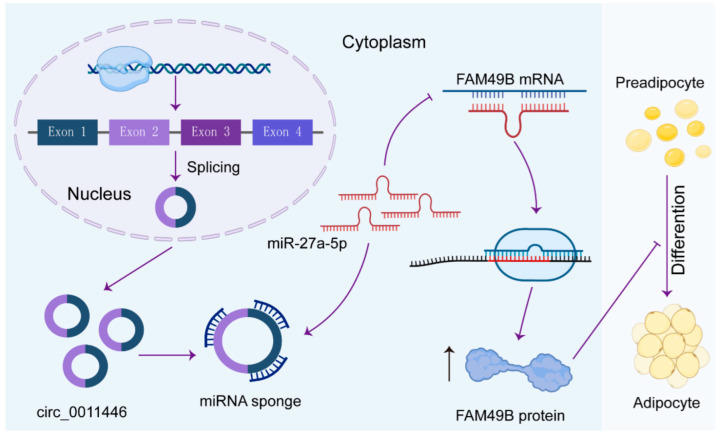
Model of circ_0011446 functioning as a ceRNA to regulate FAM49B expression by sponging miR-27a-5p, thus inhibiting the differentiation of goat intramuscular adipocyte.

**Table 1 ijms-26-02294-t001:** RNA oligonucleotides for miR-27a-5p and siRNA for FAM49B in this study.

Name	Sequence (5′-3′)
mimics NC	S: UUCUCCGAACGUGUCACGUTT
	A: ACGUGACACGUUCGGAGAATT
miR-27a-5p mimics	S: AGGGCUUAGCUGCUUGUGAGCA
	A: CUCACAAGCAGCUAAGCCCUUU
inhibitor NC	CAGUACUUUUGUGUAGUACAA
miR-27a-5p inhibitor	UGCUCACAAGCAGCUAAGCCCU
si-NC	S: UUCUCCGAACGUGUCACGUTT
	A: ACGUGACACGUUCGGAGAATT
*FAM*49*B*-Si1	S: GCAUGAGGAUUAAUAAUGUTT
	A: ACAUUAUUAAUCCUCAUGCTT
*FAM*49*B*-Si2	S: GCUACAACAAAGUUUGUAUTT
	A: AUACAAACUUUGUUGUAGCTT
*FAM*49*B*-Si3	S: GCGUCAUAAUACUCUAUGATT
	A: UCAUAGAGUAUUAUGACGCTT

**Table 2 ijms-26-02294-t002:** Related primer information.

Primer Name	Primer Sequence (5′-3′)	Product Length
divergent primer	S: GAATCGAATCCAGAAGCCAAAT A: TCCCATAGATGTTAGAAGCAAAGC	336
convergent primer	S: GCCTCCTCATTCAGCCAGC A: CAGCCCTCCATCTCTCCCA	463

**Table 3 ijms-26-02294-t003:** Related primer information.

Primer Name	Primer Sequence (5′-3′)
*U*6	S: TGGAACGCTTCACGAATTTGCG A: GGAACGATACAGAGAAGATTAGC
*GAPDH*	S: GCAAGTTCCACGGCACAG A: TCAGCACCAGCATCACCC
*UXT*	S: GCAAGTGGATTTGGGCTGTAAC A: ATGGAGTCCTTGGTGAGGTTGT
*PPAR* *Ƴ*	S: AAGCGTCAGGGTTCCACTATG A: GAACCTGATGGCGTTATGAGAC
*C/EBPα*	S: AAGCGTCAGGGTTCCACTATG A: GAACCTGATGGCGTTATGAGAC
*C/EBPβ*	S: AAGCGTCAGGGTTCCACTATG A: GAACCTGATGGCGTTATGAGAC
*SREBP*1	S: AAGCGTCAGGGTTCCACTATG A: GAACCTGATGGCGTTATGAGAC
*AP*2	S: AAGCGTCAGGGTTCCACTATG A: GAACCTGATGGCGTTATGAGAC
miR-27a-5p	S: AGTCTAAGGGCTTAGCTGCTTG
	A: GTGCAGGGTCCGAGGT
miR-27a-5p stem-loop	GTCGTATCCAGTGCAGGGTCCGAGGTATTCGCACTGGATACGACTGCTCACA
*FAM*49*B*	S: GAGGGACACGAACAGAATCACC
	A: GAGGGAGAGGAACAGAGGAAAG
*FAM*49*B* 3’UTR	S: GGAAAAGCACCTGCTGTAGAC
	A: TTTTATCATCAACAGCCATTCTT

## Data Availability

Data are contained within the article.

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
