# Peer review of "Circ_0011446 Regulates Intramuscular Adipocyte Differentiation in Goats via the miR-27a-5p/FAM49B Axis"

_ijms, 2025, doi:10.3390/ijms26052294_

Round 1

Reviewer 1 Report

Comments and Suggestions for Authors

The manuscript is well written, but need substantial revisions especially for results and discussion sections.

Comments on the Quality of English Language

The manuscript needs to be reviewed by a native speaker.

Author Response

Response to Reviewer 1 Comments

Abstract: Your abstract is well written and technically sound, but there are a few areas where clarity, grammar, and structure could be improved.

  1. Change "Here, we utilized previous sequencing data" to "In current study, we utilized existing sequencing data".

Response: Thank you for your suggestion. We have made relevant modifications.

  1. Replace "Our aim was to uncover" with "Our goal was to elucidate".

Response: Thank you for your valuable comments. We have made relevant modifications.

  1. Change "the competing endogenous RNA (ceRNA) hypothesis has identified" to "the competing endogenous RNA (ceRNA) hypothesis identifies".

Response: Thanks for your comment. We have made relevant modifications.

  1. Correct "demonstrated" to "demonstrate" in the conclusion.

Response: Thanks. Revised.

  1. Add "associated with" instead of "linked to" for a more formal tone.

Response: Thank you for your valuable comments. We have made relevant modifications.

  1. Consider adding a brief mention of the broader significance of the findings (e.g., implications for goat meat quality or livestock breeding).

Response: Thank you for your valuable comments. We have made relevant modifications.

Introduction: Your introduction is well-structured and provides a comprehensive overview of the topic. However, there are areas where clarity, grammar, and flow can be improved.

  1. Some sentences are quoted from references, which in turn are derived from other sources, it is essential to cite the original references to maintain academic integrity.

Response: We sincerely apologize for the oversight and appreciate the reviewer's attention to academic integrity. We have carefully reviewed the manuscript and ensured that all referenced content is properly cited to the original sources. The necessary corrections have been made, and we thank the reviewer for this important reminder.

  1. Change past tense (e.g., "played," "was") to present tense (e.g., "plays," "is") where appropriate, as the introduction describes general facts and ongoing research. For example: "Goat meat was among the most widely consumed meats globally" → "Goat meat is one of the most widely consumed meats globally." "IMF played a crucial role" → "IMF plays a crucial role."

Response: Thank you for your valuable comments. We have made relevant modifications.

  1. Rephrase "reasonable IMF content helps improve meat quality and flavor" to "an optimal IMF content helps improve meat quality and flavor".

Response: Thanks for your reminder. We have modified it according to your suggestions (Line 36-37).

  1. Replace "IMF was fat stored within the muscle" with "IMF refers to fat stored within the muscle".

Response: Thank you for your valuable comments. We have made relevant modifications (Line 37).

  1. Fixed spacing issues (e.g., "signaling pathways[8, 9]" → "signaling pathways [8, 9]").

Response: Thanks for your reminder. We have modified it according to your suggestions in the whole manuscript.

  1. Change "goat meat played a vital role in human nutrition" to "goat meat plays a vital role in human nutrition".

Response: Thank you for your valuable comments. We have made relevant modifications (Line 35).

  1. Rephrase "circRNAs related to adipogenesis being identified" to "circRNAs related to adipogenesis have been identified".

Response: Thanks for your comment. We have made relevant modifications (Line 63).

Materials and methods:

  1. Add details about goats (such as number, gender, diet and management).

Response: Thanks for the question about the number, gender, diet and management of animals. In this revision, we added this information in the methods. Please see line 247-250.

  1. Why did you utilize 7-day-old goats? Consider adding a brief note on the rationale for using 7-day-old goats.

Response: Thank you for your suggestion. When the intramuscular precursor adipocyte culture system of meat goats was established in the laboratory, 7-day-old goats were selected, which was adopted in the related articles published in the laboratory recently (Xiong et al., 2018; Du et al.,2021; Li et al.,2022).

[1] Xiong Y, Xu Q, Lin S, Wang Y,Lin YQ, Zhu JJ. Knockdown of LXRα inhibits goat intramuscular preadipocyte differentiation. Int J Mol Sci, 2018, 19(10): 3037.

[2] Du Y, Wang Y, Li YY, Emu QZ, Zhu JJ, Lin YQ. miR-214-5p Regulating Differentiation

of Intramuscular Preadipocytes in Goats via Targeting KLF12. Front Genet, DOI: 10.3389/ fgene.2021.748629.

[3] Li X, Zhang H, Wang Y, Li YY, Wang YL, Zhu JJ, Lin YQ. Chi-Circ_0006511 Positively Regulates the Differentiation of Goat Intramuscular Adipocytes via Novel-miR-87/CD36 Axis. Int J Mol Sci,2022, doi:10.3390/ijms232012295.

  1. The number of biological and technical replicates should be mentioned to obtain statistically significant results.

Response: Thanks for your reminder. We have modified it according to your suggestions (Line 333-335).

  1. Consider adding a brief note on the rationale for using UXT, U6, and GAPDH as internal reference genes.

Response: Thank you for your valuable comments. Based on their stable expression across various experimental conditions in adipogenesis and related biological processes, UXT, U6, and GAPDH were selected as internal reference genes. Widely used in adipocyte differentiation studies, UXT demonstrates consistent expression levels. Commonly employed for normalizing small non-coding RNAs, such as microRNAs, U6 exhibits stable expression in cellular contexts. Extensively validated in similar studies, GAPDH is a well-established housekeeping gene with stable expression in multiple cell types, including preadipocytes. Ensuring accurate normalization and reliable quantification of gene expression, these reference genes were utilized in our experiments. This description was added line 296-302.

  1. A paragraph about the Ethics consideration should be added.

Response: Thanks for your reminder. In this revision, we added this information in the methods. Please see line 247-248.

  1. P values should be in italic.

Response: Thanks for your suggestion. We have modified it according to your suggestions in the whole manuscript.

  1. Statistical analysis and experimental design is not clear.

Response: Thanks for your suggestion. We have clarified that statistical comparisons were performed using Student's t-test or one-way ANOVA with SPSS 20.0 software, and that P values < 0.05 were considered statistically significant. Additionally, we have emphasized that the results are presented as means ± SEM and are representative of three biological and three technical replicates to ensure the reproducibility and reliability of the data.

Results:

  1. You stated that “The results showed that the expression of circ_0011446 decreased in the early stage (0-72 h), and then increased again from 96 h to 120 h” (Figure 1C). Figure 1C shows that the expression of circ_0011446 decreased in the early stage to 24h and stabilized from 24h to 72h, then increased to 96h and decreased again until the end (120h).

Response: We appreciate the reviewer’s careful observation. In our statement, the expression trend of circ_0011446 was described relative to the baseline (0 h). Specifically: From 0 h to 72 h, the expression of circ_0011446 decreased significantly compared to 0 h; From 96 h to 120 h, the expression increased again relative to the lowest point at 72 h. We acknowledge that the expression stabilized between 24 h and 72 h and slightly decreased after 96 h, but the overall trend aligns with our description when compared to 0 h. We have clarified this in the revised manuscript to avoid any confusion (Line 82-86). Thank you for your valuable feedback.

  1. Again, you stated that “We observed that the expression of circ_0011446 was lowest during the 0-72 h period of goat intramuscular adipocyte differentiation, suggesting that it inhibits the early differentiation of intramuscular preadipocytes (IMPA)” and that is not right.

Response: We thank the reviewer for their observation. To clarify, the expression of circ_0011446 was indeed lowest during the 0–72 h period of goat intramuscular adipocyte differentiation compared to the baseline (0 h). This suggests that circ_0011446 may play a role in inhibiting the early differentiation of intramuscular preadipocytes (IMPA). We have revised the text to ensure this point is clearly articulated (Line 95-98).

  1. C/EBPβ expression was significantly downregulated in both overexpression and knockdown experiments; you need to give a clear discussion for this phenome.

Response: We appreciate the reviewer’s insightful comment regarding the downregulation of C/EBPβ in both overexpression and knockdown experiments. This observation suggests a complex regulatory mechanism involving circ_0011446 and its potential interaction with C/EBPβ. We have added a detailed discussion to address this phenomenon in the revised manuscript. Please see line 110-119.

  1. I think the results will be more accuracy if the experiments performed after 24 h.

Response: We sincerely thank the reviewer for their valuable suggestion regarding the timing of the experiments. We agree that 24 h is an important time point in adipocyte differentiation. However, in our study, we chose to perform the experiments at 72 h because this time point represents a critical stage in goat intramuscular adipocyte differentiation, characterized by significant lipid droplet accumulation and optimal differentiation status the intramuscular adipocytes of 3 days of induction differentiation were in a differentiated state through morphological staining and OD value measurement.  This allowed us to better capture the functional role of circ_0011446 in adipogenesis.

  1. No need to mention the non-significant differences ‘such as AP2 expression showed no significant difference”.

Response: Thanks for your comment. We have made relevant modifications.

  1. P values should be in italic.

Response: Thank you for your valuable comments. We have made relevant modifications.

Discussion: The discussion section needs fundamental improvements. The authors should summarizing, discussing, and interpreting the most important results.

  1. Information about FAM49B, miR-27a-5p, and circ_0011446 should be added and discussed.

Response: We thank the reviewer for this valuable suggestion. In response, we have added detailed information and discussion about FAM49B, miR-27a-5p, and circ_0011446 in the revised manuscript, including their roles and interactions in the regulatory network. Please see in discussion part.

  1. Many parts of the unclear results need to be discussed.

Response: Thank you for your valuable comments. We have made relevant modifications.

  1. Include a discussion on the potential biological roles, function, pathways, mechanism and implications of FAM49B, miR-27a-5p, and circ_0011446 axis.

Response: Thank you for your insightful suggestion. We have now included a discussion on the potential biological roles, functions, pathways, mechanisms, and implications of the FAM49B, miR-27a-5p, and circ_0011446 axis in the revised manuscript. We hope this addition provides a clearer understanding of the topic.

  1. The possible limitations and future perspective need to be discussed.

Response: Thanks for you suggests. In this revision, we added this information in the discussions. Please see line 286-295.

  1. Conclusion: please mention conclusion, recommendation, limitation and future studies. The arrows in Figure 7 should be reviewed.

Response: Thank you for your valuable comments. We have revised the conclusion to include a summary of key findings, recommendations, limitations, and directions for future studies.

Reviewer 2 Report

Comments and Suggestions for Authors

Wang et al. used gain of function and knock of function experimental design to test the function and underlying mechanisms of ceRNA FAM49B in the distribution of intramuscular fat (IMF) in goats preadipocytes. 

  1. Fig 1b, please indicate the individual band size
  2. Fig. 1c, please show all the data points and indicate the N number and sources.
  3. line 92, 24-72h is more accurate than 0-72h.
  4. Fig. 2A, individual data points,  also, what is the unit of the y-axis? Fold?
  5. Fig. 2B, scale bars needed, individual data plots for the graph
  6. Fig. 2c, scale bars needed. Also individual data plots needed
  7. Fig. 2d, individual data plots needed for each column, also indicate N number
  8. Fig. 2f, scale bars and individual data plots for each sample
  9. Fig. 2g, scale bars, also how you measure the fluorescent intensity w/o bias.
  10. Fig. 2H, individual data plots, also units of y-axis is missing.
  11. Fig. 3A, you need high resolution+magnification representative images for this quantification.
  12. Fig. 3D, for NC group what sample do you use? Also how many replicates are in each group?
  13. Fig. 3E and line 131, I guess you made some mistakes here describing the result.
  14. Fig. F, individual data plots needed
  15. For all fig 4, include individual data plots for graphs and make scale bars bigger.
  16. Fig. 5A, include potential binding site motif
  17. Fig. 4, you might need further experiments to test the binding between miR-27a-5g

Comments on the Quality of English Language

It would be great if proofreading could be done by professional academic writers, especially for the flow and connections between sentence.

Author Response

Response to Reviewer 2 Comments

Wang et al. used gain of function and knock of function experimental design to test the function and underlying mechanisms of ceRNA FAM49B in the distribution of intramuscular fat (IMF) in goats preadipocytes.

  1. Fig 1b, please indicate the individual band size

Response: Thanks for your reminder. In this revision, we added this information in the results. Please see Figure 1b.

  1. Fig. 1c, please show all the data points and indicate the N number and sources.

Response: Thank you for your valuable comments. We have made relevant modifications. Please see Figure 1c.

  1. line 92, 24-72h is more accurate than 0-72h.

Response: Thanks for your suggestion. We have revised it according to your suggestions. Please see results 2.2.

  1. Fig. 2A, individual data points, also, what is the unit of the y-axis? Fold?

Response: We thank the reviewer for their careful attention to the details of Fig. 2A. As noted by the reviewer, individual data points are already included in Fig. 2A to ensure transparency and reproducibility of the results. The y-axis represents the relative expression levels of circ_0011446 mRNA. We have now clarified this in the revised figure legend to avoid any ambiguity. We appreciate the reviewer's feedback and hope these clarifications address their concerns. Please find the updated Fig. 2A and its legend in the revised manuscript.

  1. Fig. 2B, scale bars needed, individual data plots for the graph

Response: We thank the reviewer for their feedback on Fig. 2B. We have added scale bars to improve data interpretation. Individual data points are already included. These changes have been incorporated into the updated Fig. 2B. We hope this addresses the reviewer's concerns.

  1. Fig. 2c, scale bars needed. Also individual data plots needed

Response: We thank the reviewer for their feedback on Fig. 2C. We have added scale bars to improve clarity. Individual Data Plots are already included, these changes have been incorporated into the updated Fig. 2C.

  1. Fig. 2d, individual data plots needed for each column, also indicate N number

Response: Thanks for your suggestion. We have now added individual data points for each column to ensure transparency. The sample size (N) for each group has been clearly indicated in the Fig. 2D. These changes have been incorporated into the updated Fig. 2D.

  1. Fig. 2f, scale bars and individual data plots for each sample

Response: Thanks for your reminder. Regarding Fig. 2f, we have now included scale bars to improve the clarity of the images. Additionally, we have provided individual data plots for each sample to allow for a more detailed examination of the data. We hope these revisions address your concerns.

  1. Fig. 2g, scale bars, also how you measure the fluorescent intensity w/o bias.

Response: Thanks for your comment. We have added scale bars to the figure to provide clear reference points for the data. Please see in update Fig. 2g. To ensure unbiased measurement of fluorescent intensity, we used ImageJ tool with standardized settings for all images. Background fluorescence was subtracted, and the intensity was normalized to a control condition to minimize variability. Detailed methods are now included in the Materials and Methods section (Line 332-336).

  1. Fig. 2H, individual data plots, also units of y-axis is missing.

Response: We thank the reviewer for their careful review of Fig. 2H. We have now added individual data points to ensure transparency and reproducibility. The y-axis represents mRNA expression levels of adipogenic marker genes following circ_0011446 knockdown (H), as clarified in the updated figure legend. These changes have been incorporated into the revised Fig. 2H.

  1. Fig. 3A, you need high resolution+magnification representative images for this quantification.

Response: Thanks for your comment. We have modified it according to your suggestions. Please see in update Fig. 3A.

  1. Fig. 3D, for NC group what sample do you use? Also how many replicates are in each group?

Response: Thanks for your reminder. The NC group uses samples overexpressing circ_0011446 to assess its effect on miR-27a-5p expression. Each group includes three biological replicates and two technical replicates to ensure the reliability of the results.

  1. Fig. 3E and line 131, I guess you made some mistakes here describing the result.

Response: Thanks for your comment. We have modified it according to your suggestions. Please see in Line 143.

  1. Fig. 3F, individual data plots needed.

Response: Thanks for your reminder. We have now added individual data points to the graph to ensure transparency and reproducibility. These changes have been incorporated into the updated Fig. 3F.

  1. For all fig 4, include individual data plots for graphs and make scale bars bigger.

Response: Thanks for your suggestion. We have now added individual data points and make scale bars bigger to the graph. These changes have been incorporated into the updated Fig. 3.

16.Fig. 5A, include potential binding site motif.

Response: We thank the reviewer for their insightful suggestion regarding Fig. 5A. While we agree that including the potential binding site motif would enhance the figure, the current data does not allow us to definitively identify the binding site. we would like to clarify that our current study focuses on predicting downstream target genes of miR-27a-5p using TargetScan, MIRDB, Starbase, and our laboratory sequencing results. However, we have not experimentally validated the binding site motifs, which limits our ability to include them in Fig. 5A. We acknowledge this as a limitation and plan to address it in future studies through experimental validation. Please see line 181-186 for a detailed description.

  1. Fig. 4, you might need further experiments to test the binding between miR-27a-5g.

Response: Thanks for your suggestion. In response, we would like to clarify that the relationship between miR-27a-5p and circ_0011446 or its target genes has been preliminarily validated using qPCR and dual-luciferase reporter assays, as described in our results (Fig 3 and Fig 5). While these experiments provide initial evidence, we acknowledge that further detailed binding studies (e.g., RIP, EMSA, or additional functional assays) would strengthen our findings. We plan to address this in future work to provide more comprehensive validation. Thank you for your valuable feedback again, which will help guide our ongoing research.

Round 2

Reviewer 1 Report

Comments and Suggestions for Authors

I suggest more revision for materials and methods and discussion sections.

If you can increase the number of samples to 6 with three replication that would be even better, but it depends on the journal's rules regarding the number of samples.

Author Response

Response to Reviewer 1 Comments

Introduction:

  1. Fixed spacing issues (e.g.,"after induced differentiation[19]" to "after induced

differentiation [19]"), please check all references.

Response: Thank you for your suggestion. We have made relevant modifications (Line 67).

Materials and methods:

  1. Add details about goats (such as gender, diet and management).

Response: Thank you for your valuable comment. In our study, we focused on 7-day-old lambs that were still in the pre-weaning stage. At this early age, the lambs were not yet separated by gender, as gender-specific management practices are typically implemented later in their development. The diet of the lambs consisted solely of maternal milk, as they had not yet been introduced to solid feed or any supplementary diet. Management during this period was primarily focused on ensuring proper maternal care and monitoring the health of the lambs, as they were entirely dependent on their mothers for nutrition and survival. All adult and lamb goats were fed and managed under standard environmental and nutritional conditions. In this revision, we added this information in the methods. Please see line 307-308.

  1. Why did you utilize 7-day-old goats? Consider adding a brief note on the rationale for using 7-day-old goats.

Response: Thanks for your comment. In our study, we utilized 7-day-old goats, as preliminary experiments had shown that a higher yield of pre-adipocytes could be isolated from kids aged 7 to 14 days. This finding is consistent with those reported in other species, such as pigs [1] and chickens [2], where younger animals also yield more proliferative and differentiation-competent preadipocytes. Additionally, at this age, the animals are still in the pre-weaning stage, and the tissue is less influenced by external factors such as diet or environmental changes, ensuring a more standardized starting material for cell isolation. In summary, this description was added line 302-308.

[1] Chen Y J, Liu H Y, Chang Y T, et al. Isolation and differentiation of adipose-derived stem cells from porcine subcutaneous adipose tissues[J]. Journal of Visualized Experiments: Jove, 2016 (109): 53886.

[2] Cherry J A, Swartworth W J, Siegel P B. Adipose cellularity studies in commercial broiler chicks[J]. Poultry Science, 1984, 63(1): 97-108.

  1. The number of biological (n=3) and technical replicates (n=2) is the minimum that should be considered for statistical analysis, I advise you to increase it to 6 biological with 3 technical replicates.

Response: We appreciate your invaluable suggestion. We sincerely apologize for any lack of clarity in our initial description of the experimental design. In our study, pre-adipocytes were isolated from three 7-day-old goats and combined to minimize individual variation. Subsequently, we conducted three biological replicates, each using independently prepared cell samples from the combined population, and two technical replicates for each biological replicate. Although this design facilitates statistical analysis, we acknowledge that increasing the number of biological replicates (e.g., to n=6) would further strengthen the robustness of our results. In future studies, we will incorporate your suggestion to incorporate more biological replicates. For the current study, we will clearly delineate the experimental design in the revised manuscript. Please see line 308-310.

  1. Consider adding a brief note on the rationale for using UXT, U6, and GAPDH as internal reference genes, references is needed.

Response: Thanks for your reminder. Based on their stable expression across various experimental conditions in adipogenesis and related biological processes, UXT, U6, and GAPDH were selected as internal reference genes. Widely used in adipocyte differentiation studies, UXT demonstrates consistent expression levels. Commonly employed for normalizing small non-coding RNAs, such as microRNAs, U6 exhibits stable expression in cellular contexts. Extensively validated in similar studies, GAPDH is a well-established housekeeping gene with stable expression in multiple cell types, including preadipocytes. Ensuring accurate normalization and reliable quantification of gene expression, these reference genes were utilized in our experiments. This description was added line 352-357.

Results:

  1. You stated that “The results showed that the expression of circ_0011446 decreased in the early stage (0-72 h), and then increased again from 96 h to 120 h” (Figure 1C). Figure 1C shows that the expression of circ_0011446 decreased in the early stage to 24h and stabilized from 24h to 72h, then increased to 96h and decreased again until the end (120h).

Response: Thank you for your careful review and valuable feedback regarding the description of the results in Figure 1C. We appreciate your attention to detail and agree with your observation. We have revised the statement in the manuscript to accurately reflect the trends shown in Figure 1C. The updated text now reads:“The results showed that the expression of circ_0011446 decreased in the early stage (0-24 h), stabilized from 24 h to 72 h, then increased at 96 h, and decreased again until the end (120 h).” Please see in Line 82-84.

  1. C/EBPβexpression was significantly downregulated in both overexpression and knockdown experiments; you need to give a clear discussion for this phenome, transfer it to discussion section and added references.

Response: Thanks for your comment. In this revision, we added this information in the discussion section. Please see line 243-247.

  1. I think the results will be more accuracy if the experiments performed after 24 h.

Response: Thanks for this comment. Our choice of 72 hours is based on several considerations. The aim of our experiment is to determine the regulatory role of circ_0011446 in intramuscular adipocyte differentiation, encompassing both undifferentiated and differentiated states. Previous lab studies identified 72 hours as a critical stage for goat intramuscular adipocyte differentiation, characterized by significant lipid droplet accumulation and optimal differentiation. Morphological staining and OD value measurement confirmed differentiation in adipocytes induced for 72 hours, allowing us to capture circ_0011446's functional role in adipogenesis. Consistency in research was ensured by utilizing the same 72-hour time point in our previous transcriptome analysis, facilitating direct result comparison. Future studies will consider additional time points to further validate and expand our findings.

  1. No need to mention the non-significant differences ‘such as AP2 expression showed no significant difference”.

Response: Thanks for your comment. We have made relevant modifications.

  1. P values should be in italic.

Response: Thank you for your valuable comments. We have made relevant modifications.

Discussion: The discussion section needs more improvements. The authors should summarizing, discussing, and interpreting the most important results.

Response: Thank you for your valuable comments. We have thoroughly revised the discussion section to better highlight the key findings. Please see line 208-298.